# Socio-spatial equity analysis of relative wealth index and emergency obstetric care accessibility in urban Nigeria
Kerry L. M. Wong [1,17], Aduragbemi Banke-Thomas [1,2,3,17] ✉, Tope Olubodun[4], Peter M. Macharia [5,6,7], Charlotte Stanton [8] ✉, Narayanan Sundararajan[8], Yash Shah[8], Gautam Prasad[8], Mansi Kansal[8], Swapnil Vispute[8], Tomer Shekel[8], Olakunmi Ogunyemi[9], Uchenna Gwacham-Anisiobi[10], Jia Wang[11], Ibukun-Oluwa Omolade Abejirinde[12,13], Prestige Tatenda Makanga[14,15], Bosede B. Afolabi[3,16] & Lenka Beňová[5]

## Abstract

**Background** Better geographical accessibility to comprehensive emergency obstetric care (CEmOC) facilities can significantly improve pregnancy outcomes. However, with other factors, such as affordability critical for care access, it is important to explore accessibility across groups. We assessed CEmOC geographical accessibility by wealth status in the 15 most-populated Nigerian cities.
**Methods** We mapped city boundaries, verified and geocoded functional CEmOC facilities, and assembled population distribution for women of childbearing age and Meta's Relative Wealth Index (RWI). We used the Google Maps Platform's internal Directions Application Programming Interface to obtain driving times to public and private facilities. City-level median travel time (MTT) and number of CEmOC facilities reachable within 60 min were summarised for peak and non-peak hours per wealth quintile. The correlation between RWI and MTT to the nearest public CEmOC was calculated.
**Results** We show that MTT to the nearest public CEmOC facility is lowest in the wealthiest 20% in all cities, with the largest difference in MTT between the wealthiest 20% and least wealthy 20% seen in Onitsha (26 vs 81 min) and the smallest in Warri (20 vs 30 min). Similarly, the average number of public CEmOC facilities reachable within 60 min varies (11 among the wealthiest 20% and six among the least wealthy in Kano). In five cities, zero facilities are reachable under 60 min for the least wealthy 20%. Those who live in the suburbs particularly have poor accessibility to CEmOC facilities.
**Conclusions** Our findings show that the least wealthy mostly have poor accessibility to care. Interventions addressing CEmOC geographical accessibility targeting poor people are needed to address inequities in urban settings.

## Plain language summary

Access to critical obstetric care can be lifesaving for pregnant women and their offspring. However, socioeconomic factors are known to affect accessibility to health services across different groups. Here, we assessed peak and off-peak travel times to functional health facilities for women from 15 Nigerian cities, using travel time estimates produced by Google Maps and stratified by wealth status. Travel time to the nearest hospital and the number of hospitals reachable within 60 min varied across cities. The wealthiest 20% across all cities had the shortest travel time and vice versa for the least wealthy 20%. Women who live in the suburbs particularly have poor accessibility. Tailored action is needed to improve access for vulnerable populations living in urban settings.

Women with complications of pregnancy and childbirth, including haemorrhage, pre-eclampsia/eclampsia, sepsis, and abortion, require timely access to emergency obstetric care (EmOC), as any delay increases odds of poor pregnancy outcomes, including morbidity and mortality[1–3]. According to Penchansky and Thomas, 'access' to care is multi-dimensional, as it relates to availability (presence of the service), accessibility (geographic

location of the service that makes it reachable), accommodation (organisation of the service to accept users), affordability (cost of the service), and acceptability (perceived quality of care)[4]. In countries with a high maternal and perinatal morbidity and mortality burden, including many African countries like Nigeria, which contributes as much as a third of the 282,000 maternal deaths that occur annually worldwide[1,5], pregnant women face

huge challenges in accessing EmOC services. These challenges occur at a micro- (relating to the woman and her circumstances), meso- (relating to the health facility she is trying to access while in an emergency), and macro-level (relating to the way the health system is organised)[6].

In 2009, the World Health Organisation (WHO) recommended that comprehensive EmOC (CEmOC), which is the full set of nine clinical and surgical evidence-based interventions, including caesarean section and blood transfusion, should be available in hospitals no further than 2–3 h travel time for most women[7]. However, there are ongoing debates about the adequacy of this travel time benchmark, as emerging evidence in urban African settings shows that women with pregnancy complications who are referred and require 30 min of travel to reach an appropriate level of care have significantly poorer outcomes than those who travel less than 10 min to care[8]. For their babies, direct travel to care of 10 min or more significantly increases their chance of being born dead[9].

There has been a wide perception that pregnant women in rural areas are particularly disadvantaged regarding access to care compared to those in urban areas[10]. Compared to rural areas, urban areas seemingly provide better geographical accessibility to infrastructure, services, and opportunities, including those for healthcare. However, this so-called urban advantage, which urban dwellers are deemed to have over rural ones in terms of infrastructure and outcomes, appears to be diminishing as urban planning is not catching up with population growth. The United Nations (UN) projects that by 2050, 70% of the world's population will live in urban areas, and 40% of the projected additional 2.5 billion people living in urban areas worldwide will be in Africa[11]. Also, in many urban areas in Africa, though travel distances to the nearest health facilities are generally shorter, in reality, travel times to these facilities are often much longer than perceived because of poor roads, traffic congestion, haphazard urban planning, and growing informal settlements[12–15]. These issues pose particular challenges for pregnant women trying to access EmOC, especially with the current rapid rate of urbanisation, which means cities are saturating and populations are spreading into the surrounding peri-urban areas. However, the urban poor face even greater challenges, with the global community recognising at the 2016 United Nations Conference on Housing and Sustainable Urban Development (Habitat III) in Quito that they are *"excluded from access to services"*[16].

Previous research shows that poverty and long travel time to care are important determinants of health facility delivery by pregnant women in Africa, though, their influence on whether women give birth in a health facility varies within and across countries[17]. Before utilisation, a woman must be able to access the facility[18]. However, her ability to access care can also be influenced by her socioeconomic status[19]. For pregnant women in an emergency, geographical accessibility to EmOC potentially has life-or-death consequences. In terms of their pathway to care, this is also varied. Some travel directly to facilities that can provide the care they need, while others must be referred to reach such facilities[20,21]. Robust research to show relationships between travel time to care needs to be able to reflect closer-to-reality pathways to care and be linkable to spatially represented wealth index data of the population, which is commonly used as a proxy for socioeconomic status in low- and middle-income country (LMIC) settings[22]. For such research done in urban areas, they also need to capture the realities of travel in such settings. However, almost all modelled EmOC geographical accessibility studies do not incorporate urban challenges such as traffic, which leads to an underestimation of travel time[14,23,24]. To date, the lack of sufficiently high spatial resolution data for closer-to-reality travel time and population wealth index has precluded robust analysis comparing the extent of EmOC accessibility between the poor and the rich in African urban areas. In this study, in which we address this knowledge gap by assessing CEmOC geographic accessibility by population wealth index in 15 of the most-populated Nigerian cities, we find that travel time to the nearest public CEmOC facility is highest and number of functional public CEmOC facilities is lowest for the least wealthy 20% in cities. We hope that the results will help identify if and where to intervene in action to improve EmOC geographical accessibility.

## Methods
### Study setting
Nigeria is administratively divided into 36 states and a Federal Capital Territory. The states are divided into 774 local government areas (LGAs). Each state has one or two major cities, with at least 20 of the major cities across the country having a population of over half a million[25]. There has been increasing urbanisation in Nigeria, with some of its reported consequences being the movement of city populations to peripheral suburbs, expansion of informal settlements, increasing traffic, and an urban health crisis[26]. For many women living in urban areas of Nigeria, the majority will travel to EmOC facilities when in situations of emergency using motorised transport[20,27]. We conducted this study in 15 Nigerian cities, all with a projected population of at least one million in 2022 (current) or 2030 (end term of the Sustainable Development Goals (SDGs))[28]: Aba, Abuja, Benin City, Ibadan, Ilorin, Jos, Kaduna, Kano, Lagos, Maiduguri, Onitsha, Owerri, Port Harcourt, Uyo, and Warri. Overall, the 15 cities accounted for 26% of the national population in 2022 [Supplementary Table 1].

### Study design
This cross-sectional study involved the assembly of data to define extended city boundaries to include peripheral suburbs, verify CEmOC facility functionality and geographic location (latitude and longitude), and map population distribution for women of childbearing age (WoCBA) aged 15–49 years to estimate the travel time to care. Details of the methods used to collect and collate data for the study as well as estimate travel time, are described below and have been published in our Data Descriptor paper[29].

### Data assembly
**Administrative boundaries**. As the precise boundaries of the selected cities were not available for spatial analysis, we established the extended boundaries of each city (including suburbs) by the LGAs that make it up. For this, we spatially superimposed the vector file of the LGA boundaries[30], WorldPop's gridded surface of the population (at 100 m$^2$ resolution)[31], Google Maps (Alphabet, Mountain View, California), and Global Human Settlement (GHS) layers showing the gridded surfaces of urban areas[32]. For each city, we selected all LGAs with areas of higher population density than their surroundings or marked as urban or sub-urban/peri-urban in the GHS layer[33]. Discussions amongst co-authors familiar with the cities helped improve the delineation process. Within the administrative boundaries, we considered the entire area comprising level 14 S2 cells, as defined by Google. The S2 cell optimises the splitting of spherical surfaces into grids of approximately equal size. Specifically, level 14 S2 cells are approximately 600 by 600 metres[34]. This resolution was selected to balance accuracy and computational power.

**Hospitals**. A list of hospitals in the 15 cities was first extracted from the 2018 Nigeria Health Facility Registry (NHFR)[35]. Information on the facility name, ownership, location (LGA and GPS coordinates), and operational status (open or closed) were retained from the NHFR. The NHFR data was supplemented by state-specific lists such as that of the Health Facilities Monitoring and Accreditation Agency in Lagos State[36] and by data gathered from stakeholders familiar with health service provision in the other states. We removed duplicates after combining the lists and assigning unique codes to each hospital. Data on service availability was obtained through a facility functionality assessment survey conducted with health facility administrators to specifically establish facilities that were operational 24 h a day and able to conduct caesarean sections (used as a proxy for CEmOC in this study, as capacity for other EmOC services provision is usually subsumed in capacity for caesarean sections[7]). We also confirmed facility ownership (public—federal or state, or private—for-profit, not-for-profit, faith-based organisations, military, or police-owned facilities). The survey was conducted during in-person hospital visitations by trained research assistants using a short questionnaire. Data collection took place from March to August 2022. Our curated list of verified CEmOC facilities is publicly available[29,37].

**Population distribution**. We obtained the constrained version of population distribution of WoCBA at 1 km$^2$ spatial resolutions from WorldPop's open spatial demographic data portal[38]. WorldPop uses dasymetric techniques to create the gridded surface by disaggregating 2006 census data from LGAs based on weights derived from covariates such as land use, land cover, and night-time lights[32]. National estimates were projected to match United Nations Population Division 2022 estimates while adjusting for differences between rural and urban areas. Age and sex multipliers data from census and household data were then applied to the projected national estimates to derive the proportion of WoCBA nationally. Geospatial layers of various resolutions were resampled to 600 by 600 metres to match the spatial granularity of the 14 S2 cells[34].

**Relative wealth**. We utilised Meta's Relative Wealth Index (RWI) as a measure of wealth at the level of S2 cell[39,40]. This index estimates the relative wealth of the people living in each micro-region relative to others in the same country. The index is based on de-identified connectivity data, geospatial "big" data from satellites and other existing sensors, used to train a machine-learning algorithm that predicts microregional poverty. Data for Nigeria was based on a survey of 40,680 households in 899 unique survey locations (known as 'villages'). The survey included questions that assessed socioeconomic circumstances of each household. RWI values are available for small areas (also referred to as micro-regions), each of approximately 2.4 km$^2$ in size[41]. RWI for each S2 cell was taken from that of its nearest micro-region.

### Computing travel time and geographical coverage
We extracted travel times for each 14 S2 grid cell from the Google Maps Platform's internal Directions Application Programming Interface (API), which uses Machine-Learning models that leverage a range of inputs to estimate travel time[42]. The travel times were based on motorised transport from each grid centre as the origin to the nearest CEmOC facilities by ownership (public and private). Motorised transport was used as available evidence from the country showed that it was the most common means of transport to care for pregnant women in emergency[20,27,43]. For each ownership type, we derived travel times during the peak traffic scenario for weekdays 18-20 h for the off-peak traffic scenario for weekends 01–03 h. The curated dataset of closer-to-reality travel time estimates for all selected cities is publicly available[29,44].

To provide summary statistics on the geographic accessibility of CEmOC and its wealth-based equity, we used city-specific cut-off values to categorise S2 cells into quintiles—ranging from the least wealthy 20% to the wealthiest 20%, denoted as Q1, Q2, Q3, Q4, and Q5. We reported the median travel time (MTT) to the nearest CEmOC facility (1. public only, and 2. public or private), as well as the average number of reachable CEmOC facilities (1. public only, and 2. public or private), during weekday 6–8 pm by quintiles. We also compared the MTT and average number of reachable facilities between peak and off-peak travel scenarios by quintiles and by city.

Pearson correlation between RWI and travel time to the nearest public CEmOC was calculated for each city. We further grouped the S2 cells by RWI (Q1-Q2, vs Q3-Q5) and compared the percentage of S2 cells with travel time to the nearest public CEmOC longer than 60 min. We produced maps to illustrate the spatial distribution of wealth and shorter/longer travel time. Travel times for S2 cells exceeding 50 km away from their nearest target facility and those with no accessible road path to any facility were not computed; these were replaced with the maximum travel time specific to the city, traffic scenarios (peak or off-peak), and facility type (public, private, or private and private) for the purpose of correlation calculation.

We conducted analysis and visualisation as static maps with R version 4.2.0 (R Development Core Team, Auckland, New Zealand) and ArcMap (ESRI ArcGIS, Redlands, California, US). The data used are publicly available and described in detail elsewhere[29].

### Ethics statement
Ethics approval for the study was obtained from the National Health Research and Ethics Committee in Nigeria (NHREC/01/01/2007-11/04/2022) and the University of Greenwich Research and Ethics Committee (UREC/21.4.7.8). Informed consent was obtained from health facility administrators for the facility functionality assessment survey. However, Informed consent was not required for publicly available datasets on population distribution and RWI, which were secondarily used in this study.

### Reporting summary
Further information on research design is available in the Nature Portfolio Reporting Summary linked to this article.

### Results
Across the 15 cities, 132,474 S2 cells were included [Table 1 and Supplementary Figs. 1–15]. We verified the existence, functionality, and location of 2020 CEmOC facilities: ranging from 26 (Maiduguri) to 796 (Lagos).

### Table 1 | Summary characteristics of the included cities

| City | Number of S2 cells | Public hospitals | Private hospitals | All hospitals | Estimated population 2022 | Estimated population 2030 | Number of WoCBA, 2022 | Percentage of WoCBA, 2022 |
|---|---|---|---|---|---|---|---|---|
| Aba | 2337 | 2 | 106 | 108 | 1,150,116 | 1,527,000 | 377,554 | 32.8 |
| Abuja | 18,989 | 18 | 53 | 71 | 3,652,029 | 5,119,000 | 1,095,195 | 30 |
| Benin City | 16,772 | 4 | 70 | 74 | 1,841,084 | 2,451,000 | 482,055 | 26.2 |
| Ibadan | 8844 | 11 | 152 | 163 | 3,756,445 | 4,956,000 | 955,580 | 25.4 |
| Ilorin | 2965 | 7 | 68 | 75 | 1,000,477 | 1,314,000 | 283,066 | 28.3 |
| Jos | 8947 | 6 | 70 | 76 | 942,167 | 1,236,000 | 344,494 | 36.6 |
| Kaduna | 21,911 | 5 | 46 | 51 | 1,158,048 | 1,499,000 | 513,105 | 44.3 |
| Kano | 8507 | 16 | 129 | 145 | 4,219,209 | 5,551,000 | 1,294,941 | 30.7 |
| Lagos* | 11,393 | 26 | 770 | 796 | 15,387,639 | 20,600,000 | 3,402,451 | 22.1 |
| Maiduguri | 2535 | 5 | 21 | 26 | 822,337 | 1,071,000 | 265,740 | 32.3 |
| Onitsha | 2139 | 1 | 116 | 117 | 1,552,630 | 2,138,000 | 397,541 | 25.6 |
| Owerri | 780 | 2 | 74 | 76 | 945,046 | 1,282,000 | 230,314 | 24.4 |
| Uyo | 5138 | 3 | 45 | 48 | 1,264,636 | 1,771,000 | 461,254 | 36.5 |
| Port Harcourt | 11,367 | 5 | 79 | 84 | 3,324,694 | 4,595,000 | 828,146 | 24.9 |
| Warri | 9850 | 9 | 101 | 110 | 942,683 | 1,304,000 | 567,385 | 60.2 |
| Total | 132,474 | 120 | 1900 | 2020 | 41,959,240 | 56,414,000 | 11,498,821 | 27.4 |

+S2 cells with a population size 0 were excluded from the analysis. Estimates of city population for 2022 and 2030 were obtained from the UN Urbanisation Prospectus, and women of childbearing age (WoCBA) data were obtained from WorldPop project. The population of Lagos is disputed, with the State government estimating its 2022 population as 26 million.

Estimated population by 2030 ranges from 1.1 million in Maiduguri to 20.6 million in Lagos. The estimated population of WoCBA for 2022 was 11.5 million (6% of the country's overall population) [Table 1].

In 13 of 15 cities, MTT to the nearest public CEmOC facility was lowest in the wealthiest 20% of cells. In Benin City, for instance, MTT in Q5 was 43 min versus over 60 min in Q1 and Q2. The largest difference between Q1 and Q5 was observed in Onitsha (26 vs. 81 min), whilst the smallest difference was observed in Warri (20 vs. 30 min). When incorporating private CEmOC facilities (i.e., MTT to nearest public or private facility), MTT for all quintiles in all cities reduced, though the magnitudes of the reductions varied. For example, reductions were relatively mild in all groups in Maiduguri and Abuja. Nonetheless, MTT in all groups and all cities dropped below 60 min, and in seven cities, including Aba, Uyo, Kano, Ilorin, Jos, Maiduguri, and Warri, MTT in all groups dropped below 30 min [Fig. 1].

The average number of public CEmOC facilities reachable within 60 min was as high as 12 among the wealthiest 20% of cells in Kano, where reachable facilities totalled seven in Q1. In Lagos, Kaduna, Onitsha, Port Harcourt, and Benin City, zero facilities were reachable under 60 min for the least wealthy 20%. Lagos had the widest inequality gap for reachable public CEmOC facilities (nine in Q5 vs zero in Q1), followed by Ibadan and Kano. The top-inequality pattern (the wealthiest subgroup experiencing considerably better accessibility than other groups) appeared in Abuja, Jos, and

Port Harcourt. With additional consideration of private CEmOC facilities, the number of reachable facilities exceeded 30 for all groups, including Q1, in Ibadan, Ilorin, Uyo, Kano, Owerri, and Aba. Lagos remained with the widest inequality gap, where 4 and 293 facilities were within reach under 30 min for the lowest and highest wealth quintiles, respectively [Fig. 1].

As a general pattern, MTT and the number of reachable facilities during peak and off-peak traffic were similar in most cities and wealth groups. However, some exceptions were noted. In Lagos, for instance, the number of reachable facilities within 60 min was nine during peak traffic and 16 during off-peak traffic in the wealthiest 20%. In Abuja, the number of reachable facilities during peak and off-peak traffic was eight and 11, respectively [Fig. 2].

All cities had negative correlation coefficients between MTT to the nearest public CEmOC and RWI. The correlation coefficient was the largest in magnitude in Onitsha ($r = -0.70$, 95% CI [$-0.73, -0.68$]) and smallest in Aba ($r = -0.26$, 95% CI [$-0.30, -0.22$]) [Fig. 3], where the percentages of the least wealthy 40% of S2 cells that had MTT above 60 min were 86.1% and 0.3%, respectively [Table 2]. In Onitsha, the locations of such cells spanned the southern part, whilst the MTT was within 60 min in the North, where RWI was generally higher [Fig. 4]. Alongside Onitsha, three other cities had over 60% poorer S2 cells, mostly in the city outskirts, with longer MTT of 60+ minutes (Port Harcourt, Kaduna, and Benin City). In Warri,

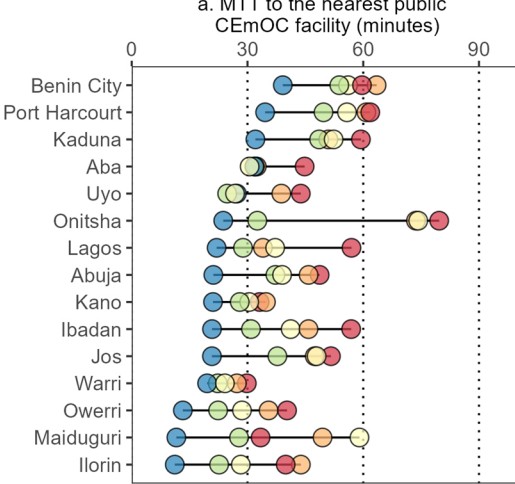

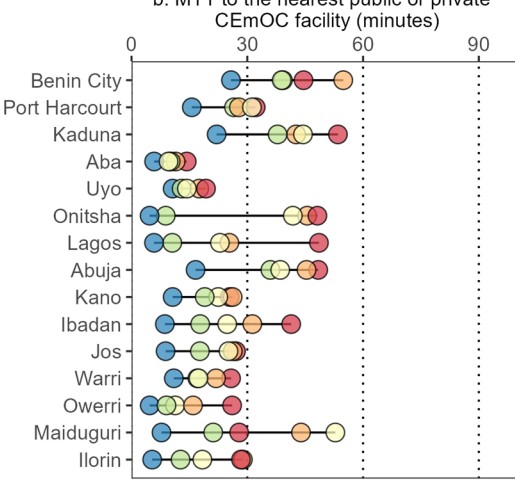

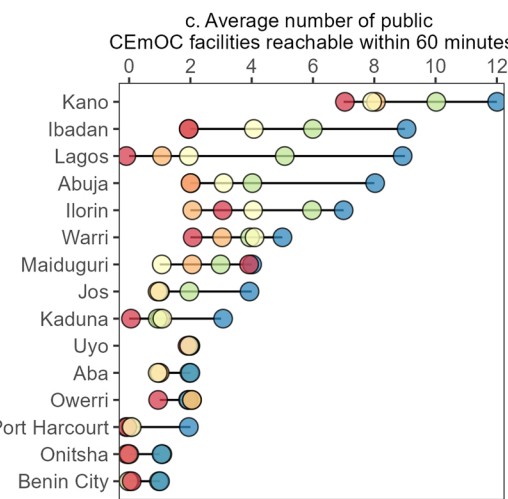

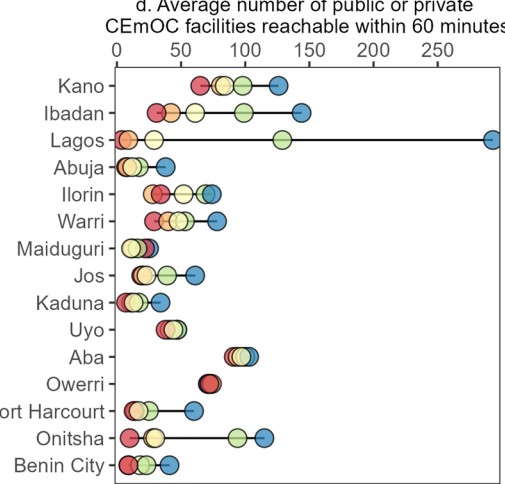

**Fig. 1 | Equiplot of geographic accessibility by relative wealth in 15 cities in Nigeria. a** Median Travel Time (MTT) to the nearest public comprehensive emergency obstetric care (CEmOC) facility in minutes in 15 cities in Nigeria by quintile of relative wealth. **b** MTT to the nearest public or private CEmOC facility in minutes.

**c** Average number of public CEmOC facilities reachable within 60 min. **d** Average number of public or private CEmOC facilities reachable within 60 min. In all panels, red circles correspond to the least wealth 20% of S2 cells in that city (Q1), orange (Q2), yellow (Q3), and green (Q4), and blue circles correspond to the wealthiest 20% of S2 cells.

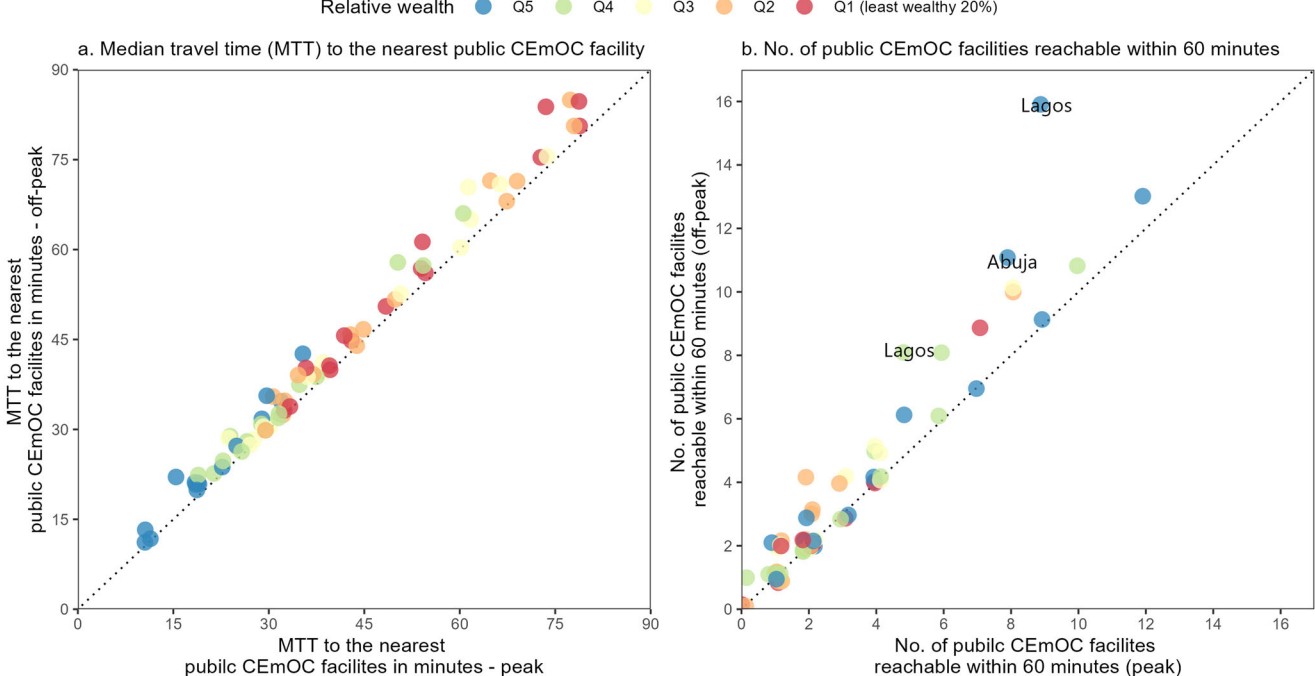

**Fig. 2 | Geographic accessibility to the nearest public CEmOC facility in 15 cities in Nigeria by wealth quintile of S2 cells—peak vs. off-peak. a** Median travel time (MTT) to the nearest public comprehensive emergency obstetric care (CEmOC) facility during peak hours (weekday 18–20 h) and off-peak hours (weekend 01-03 h) shown as a scatter plot, with each circle corresponding to a wealth quintile in one of 15 cities in Nigeria. **b** Number of public CEmOC facilities reachable within 60 min

during peak hours (weekday 18–20 h) and off-peak hours (weekend 01-03 h) shown as a scatter plot, with each circle corresponding to a wealth quintile in one of 15 cities in Nigeria. In both panels, red circles correspond to the least wealth 20% of S2 cells in that city (Q1), orange (Q2), yellow (Q3), green (Q4), and blue circles correspond to the wealthiest 20% of S2 cells.

Maiduguri, Uyo, Ilorin, Kano, Owerri, and Aba, over 80% of the least wealthy 40% of cells had MTT ≤ 60 min [Fig. 3].

## Discussion

This study assessed CEmOC geographical accessibility by population wealth index in 15 cities in Nigeria using closer-to-reality travel time estimates during peak and off-peak traffic times and to public and public or private facilities. Our study found that in 13 of 15 cities, MTT to the nearest public CEmOC facility was lowest in areas habited by the wealthiest 20% of the population. Similarly, the number of functional public CEmOC facilities that could be reached was typically the lowest amongst the poorest 20% of the urban population across cities, with the least wealthy 20% in some cities unable to reach one public facility within 60 min. The number of accessible facilities varied in some cities depending on travel periods (peak v. non-peak), and correlation coefficients between MTT to the nearest public CEmOC and RWI were negative in all cities. Broadly, it appeared those who lived in the core center of the city had better accessibility to functional public CEmOC facilities, with a higher proportion of those with better accessibility being the wealthiest 60%.

Our study revealed strong within-city inequities for CEmOC accessibility aligned with the RWI of places where urban dwellers are domiciled. The least wealthy 20% travel furthest to functional public CEmOC. We observed the largest difference in MTT between the wealthiest 20% and poorest 20% in Onitsha (26 vs 81 min) and the smallest difference in Warri (20 vs 30 min). Similarly, the average number of public CEmOC facilities reachable within 60 min varied across cities. It was as high as 11 among the wealthiest 20% in Kano, while reachable facilities totalled six amongst the least wealthy. In some cities, including Lagos, Kaduna, Onitsha, Port Harcourt, and Benin City, zero facilities were reachable under 60 min for the least wealthy 20%. Reviewing the spatial distribution of where the least wealthy population live compared with the areas that have poor accessibility to functional public CEmOC facilities, it becomes clear that those who live in

the suburbs are particularly vulnerable. This aligns with findings from a previous assessment of geographical accessibility, which recreated journeys of pregnant women who presented in an emergency in Lagos[45]. In our study, correlation coefficients between MTT to the nearest public CEmOC and RWI were negative in all cities, suggesting lower MTT from wealthier places.

Some additional insights that can be gleaned from our study include that geographical accessibility, looking at peak and off-peak travel times, were similar for most cities. However, larger cities, such as Abuja, Kano, and Lagos, generally had worse accessibility in terms of the number of functional public CEmOC facilities that could be reached during peak traffic times. In a separate study, we showed that many slum areas, where many of the least wealthy populations are domiciled, have the worst accessibility to public sector care[46]. A previous study in Bangladesh found that variability in traffic congestion significantly affected travel time to care and availability of healthcare services for slum populations, with only 63% of the city's slum population able to reach emergency services within 60 min[24].

The inequities observed somewhat relate to the distribution of the facilities, with more functional ones situated more in the core of the cities than in the peripheral suburbs. For example, in Kano, where almost the entire city population could reach functional public CEmOC facilities within 60 min, there was a relatively even distribution of facilities across space. Although adding the private facilities reduced travel time and massively increased the number of available facilities within 60 min of travel for many cities, there remains a cost barrier in accessing care in such facilities for the least wealthy that needs to be considered[12,47,48]. In addition, concerns have been raised about the quality of EmOC available within the private sector[49]. Irrespective of the potential opportunity that private facilities bring to the health system, governments must continue strategically exploring opportunities to situate CEmOC facilities in areas of geographical access inequities. By leveraging evidence such as the one we have provided in this study, which can also be presented in a digital dashboard, policymakers will be able to identify optimal location for such facilities that guarantee the best

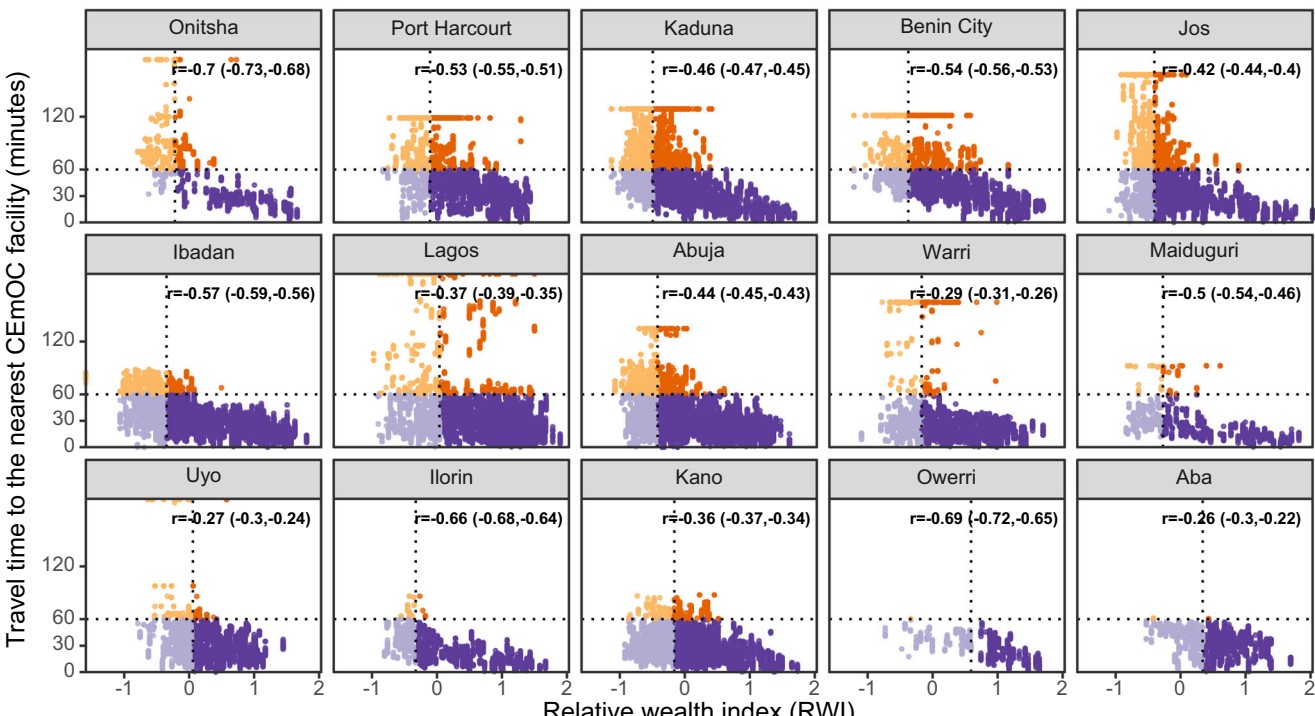

**Fig. 3 | Relative wealth index and travel time of individual S2 cells in 15 cities in Nigeria (weekday 18–20 h).** The relative wealth index (RWI) and travel time to the nearest CEmOC facility of individual S2 cells in 15 cities in Nigeria are shown as a scatter plot. Each S2 cell is represented by a circle, with the colour of the circle indicating 1. its relative wealth compared against other cells in the same city, and 2. travel time from the cell to the nearest public CEmOC facility. Purple for MTT ≤ 60 min; orange for MTT > 60 min.

value for money, despite the cost of service provision, can be established[50–53]. A public CEmOC facility in Lagos led to a 42-minute reduction in mean travel time in Lagos[45].

**Table 2 | Distribution of S2 cells by Relative Wealth Index and MTT to the nearest CEmOC facility (%)**

| | Least wealthy 40% of S2 cells | | Wealthiest 60% of S2 cells | |
|---|---|---|---|---|
| | MTT > 60 min | MTT ≤ 60 min | MTT > 60 min | MTT ≤ 60 min |
| Aba | 0.3 | 99.7 | 0.2 | 99.8 |
| Abuja | 27.5 | 72.5 | 7.5 | 92.5 |
| Benin City | 53.0 | 47.0 | 22.4 | 77.6 |
| Ibadan | 33.8 | 66.2 | 5.5 | 94.5 |
| Ilorin | 6.9 | 93.1 | 0.3 | 99.7 |
| Jos | 46.3 | 53.7 | 16.0 | 84.0 |
| Kaduna | 57.9 | 42.1 | 30.8 | 69.2 |
| Kano | 5.3 | 94.7 | 3.3 | 96.7 |
| Lagos | 29.0 | 71.0 | 5.4 | 94.6 |
| Maiduguri | 13.3 | 86.7 | 6.7 | 93.3 |
| Onitsha | 86.1 | 13.9 | 23.7 | 76.3 |
| Owerri | 0.3 | 99.7 | 0.0 | 100.0 |
| Port Harcourt | 58.5 | 41.5 | 21.3 | 78.7 |
| Uyo | 11.3 | 88.7 | 1.1 | 98.9 |
| Warri | 19.1 | 80.9 | 5.6 | 94.4 |

There are several strengths of our approach worth highlighting. First, our study was based on algorithmic outputs generated using Google Maps Platform's internal Directions API, the external version of which has been shown to offer closer-to-reality time estimates[14]. Only a handful of published studies have investigated the link between wealth and healthcare utilisation in LMIC settings, and even fewer have focused on explaining the socio-demographic inequality between wealth and healthcare accessibility[15,17,24,54], with the few that looked at it, using mostly modelled travel time estimates which were less reflective of real-world travel, especially in the urban space[14]. Our analysis sheds light on the realistic geographic accessibility barriers that hindered the less wealthy more markedly; we hope this adds new insights to the evidence base of health inequality and health service planning and policy[55]. This approach can be scaled up to ensure the use of closer-to-reality travel time estimates is combined with other datasets to inform policy in urban Africa[56]. In addition, we have included only facilities verified as having the capacity to provide caesarean section, with previous analyses including all facilities and not specifically confirming service capacity. Furthermore, our approach reflected traffic variability, highlighting accessibility during peak and off-peak travel times, which is critical for a holistic understanding of accessibility in urban areas. Finally, we have used a finer spatial scale at 0.6 by 0.6 km for our analysis of travel time—an approach deemed by expert to ensure greater accuracy and robustness in model inputs for informing policy- and decision-making[57].

However, our results should be interpreted with certain limitations in mind. First, RWI was modelled on many predictors but did not include traffic. This permits our investigation into the relationship between RWI and TT. However, among the model predictors for RWI were population density, road density, and 'built-up'-ness[39]. All these play into traffic and travel time to some extent, which might mean some correlations in the

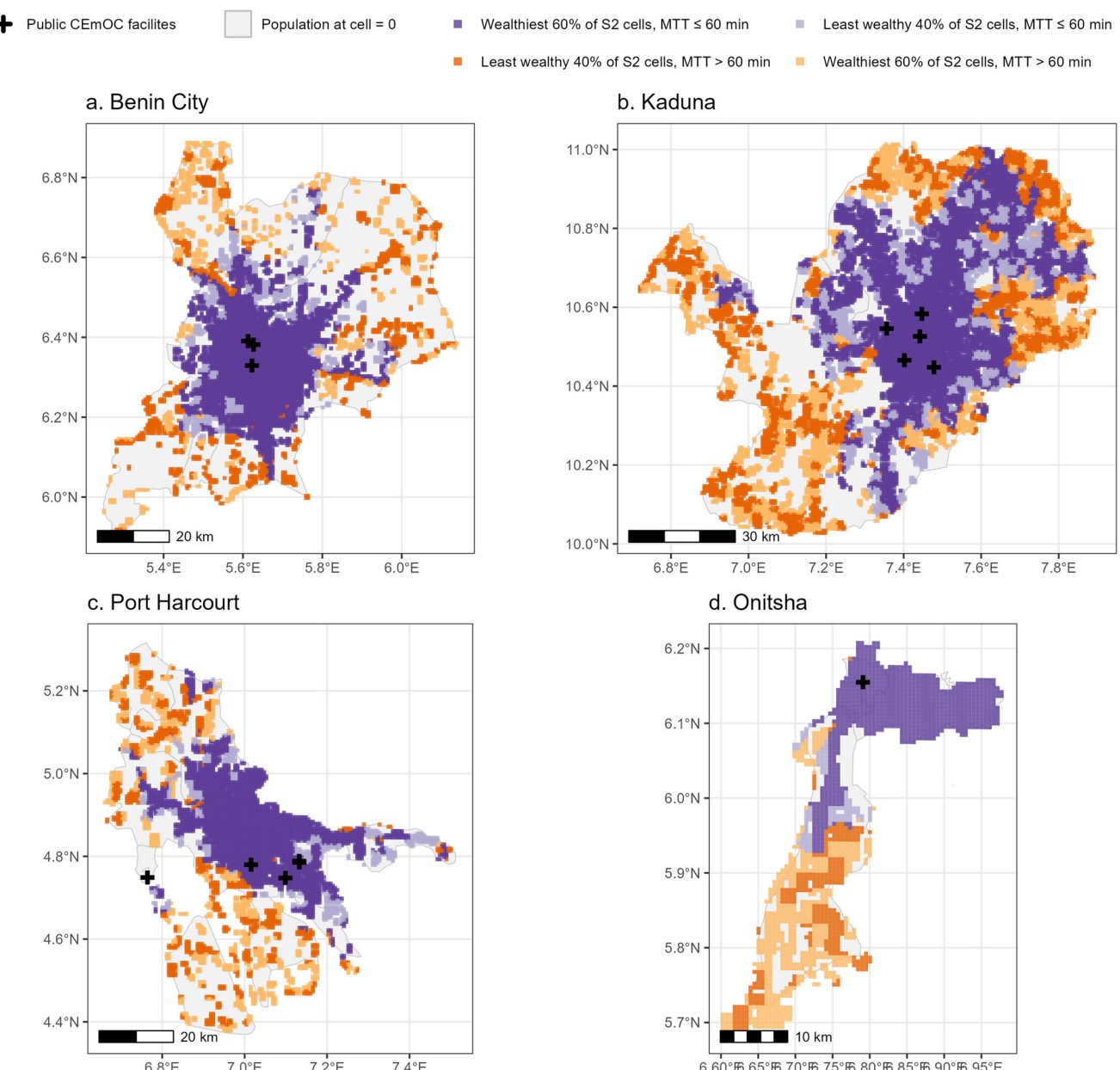

**Fig. 4 | Spatial distribution of public CEmOC and spatial distribution of relative wealth index at S2 cell level in Benin City, Kaduna, Onitsha, and Port Harcourt.** The locations of public CEmOC facilities and relative wealth index of individual S2 cells in four selected Nigerian cities (**a** Benin City, **b** Kaduna, **c** Port Harcourt, and **d** Onitsha). The locations of cells by relative wealth and median travel time (MTT ≤ 60 min or not) are also presented. Purple for MTT ≤ 60 min; orange for MTT > 60 min.

modelling of the two parameters. Second, RWI was only available at a resolution of 2.4 km², which might not be small enough to delineate a small slum within dense urban areas. Even when the granularity is 'enough', there is a risk of lack of predictive power of the included predictors to identify city slums: population density, topography-related characteristics, rainfall, road density, and mobile service usage. Third, we have assumed motorised transport as the mode of travel for our analysis and that it is available for all pregnant women when they need to commence their travel to care. While evidence suggests that the majority of pregnant women in emergency travel to facilities in some form of motorised transport, it is more likely that the least wealthy population will not have motorised transport of their own and will need additional time to get access to one before they can commence their journey to get access to care[20,27]. Fourth, we have focused our analysis only on the nearest facility despite recognising that sometimes pregnant women

bypass such facilities. In the context of the inequalities that this paper aimed to address, it made sense to focus on the nearest only, as it represents the 'best-case' of the inequalities; travel time to the second and third nearest would only be worse[44]. Fifth, travel time covering distance exceeding 50 km and water-based travels were not computed. These were replaced with context-specific maximum values, instead of omitted, in our calculation. Sixth, we took the pragmatic decision to consider caesarean section capacity as a proxy for all EmOC services in public and private facilities. However, it is possible that not all EmOC services are necessarily available at facilities with capacity to provide caesarean sections. For example, assisted vaginal delivery or blood transfusion may not be available in hospitals at any time, whether public or private[6,20]. Finally, our urban delimitation was based on a qualitative assessment, as we included areas ranging from very dense to peri-urban. While this corresponds to the expansion of urban areas towards the peripheries and

reflects the lived reality of the rapid urbanisation of the peri-urban areas in Nigeria, creating conurbations, it also means that some fringes of the selected areas may include not-so-'urban' segments.

## Conclusions

Put together, the top-inequality pattern in MTT seen in a number of cities, including Benin City, Port Harcourt, Kaduna, Abuja, and Jos, is particularly worrisome, as distribution of services is seemingly skewed towards the wealthiest places, leaving people in the remaining parts of the city (the vast majority) with little to no service. The intra-urban inequitable patterns observed in our study might not be unique to Nigeria. In many other African cities that are set up in similar ways[58], health service provision, distribution, and their spatial relationship with the characteristics of communities remain work for future research. As cities grow, geographical accessibility must be top of the agenda, especially for the particularly vulnerable poor populations moving into these urban settings, many of whom already find the high cost to access and use EmOC a huge burden[48]. The service geo-accessibility gaps identified in this study can be used to guide/prioritise attention and resources, as a matter of urgency, to the people and places most in need. While a range of interventions have been implemented in many African countries[59–61], specific interventions to address observed inequities in geographical accessibility will vary depending on local context. However, such interventions must target poor people, especially those with long travel to care places them at risk of emergency caesarean that could have been avoided and poor pregnancy outcomes[8,9,62]. Such interventions could include the construction of functional health facilities strategic stationing of ambulances in the suburbs, and transport support provided to poor pregnant women to access care in an emergency. In conclusion, we echo the assertion that "urban planning failures are evident in all cities of all sizes, leading to the growth of informal settlements and slums where the urban poor are excluded from access to services"[16]. We must ensure that we leave no space behind if we want to leave no one behind[63].

## Data availability

The main datasets used in this study can be accessed via their respective URLs or identifiers. (1) A geospatial database of close-to-reality travel times to obstetric emergency care in 15 Nigerian conurbations (https://doi.org/10.6084/m9.figshare.22699759.v1)[44] (2) Geocoded database of health facilities with verified capacity for cesarean section in urban Nigeria (https://doi.org/10.6084/m9.figshare.22689667)[37], (3) Estimates of a total number of people per grid square, adjusted to match the corresponding UNPD 2020 estimates and broken down by gender and age groupings, produced using Ecopia.AI and Maxar Technologies building footprints. (https://hub.worldpop.org/geodata/summary?id=50493)[38], and (4) Relative Wealth Index (https://data.humdata.org/dataset/relative-wealth-index)[40]. Source data which refers to the numerical values underlying Figs. 1–4 in this article are available via Supplementary Data 1.

## Code availability

All the codes/scripts used in the analyses presented this study can be found in the repository, Zenodo, and accessible via this link: https://doi.org/10.5281/zenodo.10579736[64].

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

## Acknowledgements

We would like to express our most sincere appreciation to the Nigerian Federal Ministry of Health and all state-level ministries of health involved in the project. We are also indebted to the research assistants (fifth- and sixth-year medical students from the University of Ilorin, University of Benin, University of Jos, University of Ibadan, Nnamdi Azikiwe University, University of Uyo, University of Lagos and Chukwuemeka Odumegwu Ojukwu University; a nurse from Bingham University Teaching Hospital; medical doctors from Ahmadu Bello Teaching Hospital and Lagos University Teaching Hospital; and research assistants from the states of Abia, Borno, Kano, Port Harcourt, Imo, Delta and the Federal Capital Territory) who supported the health facilities validation exercise, from 10th May 2022 to 9th August 2022. The OnTIME project led by AB-T was funded by Google. AB-T and BA were funded by the Bill and Melinda Gates Foundation (Investment ID: INV-032911). PMM was supported by Newton International Fellowship (Number NIF/R1/201418) of the Royal Society and acknowledges the support of the Wellcome Trust to the Kenya Major Overseas Programme (Number 203077). UGA is funded by a joint Clarendon/Balliol College/Nuffield Department of Population Health DPhil scholarship. LB was funded in part by the Research Foundation–Flanders (FWO) as part of her Senior Postdoctoral Fellowship.

## Author contributions

KLMW and AB-T conceptualised the study and prepared the analytical plan for the study with support from TO and LB. AB-T, TO, OO, UGA, and BBA led the facility functionality verification activity conducted as part of the study. CS, NS, YS, GP, MK, SV, and TS were involved in aggregating the travel time estimates from the Internal API used for the study. AB-T and KLMW conducted the literature review that informed the study. KLMW analysed the data with support from AB-T, PMM, and LB. AB-T and KLMW prepared the first draft of the manuscript. AB-T, KLMW, PMM, TO, JW, I-OA, PTM, BBA, and LB contributed to the interpretation of data. All authors edited the article and approved the final version of the manuscript. AB-T, KLMW, PMM, PTM, CS, NS, YS, GP, MK, SV, and TS had full access to all the data in the study, and all authors had final responsibility for the decision to submit for publication.

## Competing interests

CS, NS, YS, GP, MK, SV, and TS are employees of Google LLC, which makes the Google Maps Platform. AB-T received grant funding from Google to support this work. PMM is an Editorial Board Member for Communications Medicine and Guest Editor for the Geospatial Analysis for Improved Understanding of Health Inequalities Collection but was not involved in the editorial review or peer review, nor in the decision to publish this article. All other authors declare no competing interests.

## Additional information

[1]Faculty of Epidemiology and Population Health, London School of Hygiene & Tropical Medicine, London, UK. [2]School of Human Sciences, University of Greenwich, London, UK. [3]Maternal and Reproductive Health Research Collective, Lagos, Nigeria. [4]Department of Community Medicine and Primary Care, Federal Medical Centre Abeokuta, Abeokuta, Ogun, Nigeria. [5]Department of Public Health, Institute of Tropical Medicine, Antwerp, Belgium. [6]Population & Health Impact Surveillance Group, Kenya Medical Research Institute-Wellcome Trust Research Programme, Nairobi, Kenya. [7]Centre for Health Informatics, Computing, and Statistics, Lancaster Medical School, Lancaster University, Lancaster, UK. [8]Google LLC, Mountain View, CA, USA. [9]Lagos State Ministry of Health, Ikeja, Lagos, Nigeria. [10]Nuffield Department of Population Health, University of Oxford, Oxford, UK. [11]School of Computing & Mathematical Sciences, University of Greenwich, London, UK. [12]Dalla Lana School of Public Health, University of Toronto, Toronto, Canada. [13]Women's College Hospital Institute for Health System Solutions and Virtual Care, Toronto, Canada. [14]Surveying and Geomatics Department, Midlands State University Faculty of Science and Technology, Gweru, Midlands, Zimbabwe. [15]Climate and Health Division, Centre for Sexual Health and HIV/AIDS Research, Harare, Zimbabwe. [16]Department of Obstetrics and Gynaecology, College of Medicine of the University of Lagos, Lagos, Nigeria. [17]These authors contributed equally: Kerry L. M. Wong, Aduragbemi Banke-Thomas. ✉e-mail: Aduragbemi.Banke-Thomas@lshtm.ac.uk; chstanton@google.com

