## [Peer Review File · Communications Medicine]

Reviewers' comments:

Reviewer #1 (Remarks to the Author):

Thank you for providing me with the opportunity to review this significant piece of work. The authors have effectively tackled a crucial gap in the existing literature, specifically the modeling of access to health services in urban areas. While conventional methods like network analysis or least-cost path algorithms have proven beneficial in assessing access to care in rural regions, they fall short in capturing the complexity of travel time in urban settings. These approaches tend to underestimate actual travel time and may wrongly imply that access is not a concern in urban areas.

In this paper, the authors utilize Google data to estimate the median travel time for women of childbearing age in Nigerian cities. Their findings shed light on the overlooked issue of travel time to Comprehensive Emergency Obstetric and Neonatal Care (CEmONC) in urban areas, particularly impacting women in low-wealth quintiles. I commend the authors for their important contribution to the literature and for addressing this gap.

Moving forward, I would encourage the authors to consider further strengthening four main points:

1. Population distribution: The authors mention the use of a constrained population distribution layer specifically created for women of childbearing age. However, the process of constructing this layer is not well explained. It is worth noting that WorldPop typically does not generate constrained layers for sub-groups like this, and while pregnancy data is available at a 1km² resolution, it is unconstrained. Therefore, it would be beneficial for the authors to provide further clarification within the text regarding how they obtained or produced this specific layer. Especially because population data can cause large uncertainty in accessibility estimates (Hierink, et al. 2022)
2. S2 cells: The paper should explain the meaning of S2 cells a bit earlier. It is now explained in the sub-heading "Computing travel time and geographical coverage" but this should come in the subheading "Relative wealth" for people that do not know the Google data format.
3. Rural areas: In the discussion the authors do not mention whether this method can also be useful in rural areas or whether more classical methods are still important. Can the authors shine light on this?
4. Transferability: Can the authors add some information whether the travel times observed in Nigeria and the patterns found in the google data can be transferred to urban settings in other countries?

Minor points:

1. The following sentence in the introduction is a bit unclear: "However, there are ongoing debates about the adequacy of this travel time benchmark, as emerging evidence in urban African settings shows that women with pregnancy complications who reach care even within an hour of travel have significantly poorer outcomes." Poorer outcomes than who? Compared to rural populations?
2. Figure 1 and Figure 2 could be merged and be different panels A, B, C, D
3. In the results under Figure 2 rephrase: "Lagos remained with the widest inequality gap driven by many reachable facilities from Q5 S2 cells (n=263)". I would rephrase the sentence the other way around: "Lagos remained with the widest inequality gap driven by limited facilities available for the lowest wealth quintiles"

Reviewer #2 (Remarks to the Author):

General comment - The manuscript reports a very interesting study with a novel methodology to fill the gaps in previous studies examining healthcare access and utilization issues, particularly among women seeking obstetric care. The paper is well-written and the illustrations are adequate. A few areas I would want the authors to look at are as follow;

1. The term 'socioeconomic status' is quite a broad one and encompasses other indices beside wealth. I would suggest the authors stick with the wealth index they employed in the title or add a sentence in the abstract and introduction on using wealth index as a corollary of socioeconomic status in the study, instead of using both interchangeably.
2. The sentence 'In countries with a high maternal and perinatal morbidity and mortality burden, including many African countries like Nigeria...' in the first paragraph of the Introduction section would read better if buttressed with some current data on maternal and perinatal morbidity and mortality for context.
3. The relationship between socioeconomic status and access to healthcare services is only mentioned in passing in the introductory section. The authors expanded elaborately on the aspects about travel time and access but did not do same to socioeconomic status in justifying the study objective.
4. Just like when using a DHS dataset for a study of this nature, the authors need to creatively explain the methodology in computing the relative wealth index. It is not sufficient to assume that because the data is in the public domain, everyone understands the metadata and how the index was generated. the RWI is a major explanatory variable in the study and should be amplified accordingly.
5. The discussion section looks more like a combination of an extension and elaborated results section, a further justification of the methods employed, and a limitation to the study. Except if conforming to the journal's authors guidelines, this section should be unbundled.
6. Specific to the discussion, one would have loved to see how the study findings actually deviate / align with previous works and / or expand on previous studies that examined similar objectives but did not employ the methods used here. This will accentuate the shortcomings in those studies and further justify why this study was necessary in filling the gaps.

Thank you for providing me with the opportunity to review this significant piece of work. The authors have effectively tackled a crucial gap in the existing literature, specifically the modeling of access to health services in urban areas. While conventional methods like network analysis or least-cost path algorithms have proven beneficial in assessing access to care in rural regions, they fall short in capturing the complexity of travel time in urban settings. These approaches tend to underestimate actual travel time and may wrongly imply that access is not a concern in urban areas.

In this paper, the authors utilize Google data to estimate the median travel time for women of childbearing age in Nigerian cities. Their findings shed light on the overlooked issue of travel time to Comprehensive Emergency Obstetric and Neonatal Care (CEmONC) in urban areas, particularly impacting women in low-wealth quintiles. I commend the authors for their important contribution to the literature and for addressing this gap.

Moving forward, I would encourage the authors to consider further strengthening three main points:

1. **Population distribution:** The authors mention the use of a constrained population distribution layer specifically created for women of childbearing age. However, the process of constructing this layer is not well explained. It is worth noting that WorldPop typically does not generate constrained layers for sub-groups like this, and while pregnancy data is available at a 1km² resolution, it is unconstrained. Therefore, it would be beneficial for the authors to provide further clarification within the text regarding how they obtained or produced this specific layer. Especially because population data can cause large uncertainty in accessibility estimates (Hierink, et al. 2022)
2. **S2 cells:** The paper should explain the meaning of S2 cells a bit earlier. It is now explained in the sub-heading *“Computing travel time and geographical coverage”* but this should come in the subheading *“Relative wealth”* for people that do not know the Google data format.
3. **Rural areas:** In the discussion the authors do not mention whether this method can also be useful in rural areas or whether more classical methods are still important. Can the authors shine light on this?
4. **Transferability:** Can the authors add some information whether the travel times observed in Nigeria and the patterns found in the google data can be transferred to urban settings in other countries?

Minor points:

1. The following sentence in the introduction is a bit unclear: *“However, there are ongoing debates about the adequacy of this travel time benchmark, as emerging evidence in urban African settings shows that women with pregnancy complications who reach care even within an hour of travel have significantly poorer outcomes.”* Poorer outcomes than who? Compared to rural populations?
2. Figure 1 and Figure 2 could be merged and be different panels A, B, C, D
3. In the results under Figure 2 rephrase: *“Lagos remained with the widest inequality gap driven by many reachable facilities from Q5 S2 cells (n=263)”*. I would rephrase

the sentence the other way around: "Lagos remained with the widest inequality gap driven by limited facilities available for the lowest wealth quintiles"

Reviewers' comments:

Reviewer #1:

Thank you for providing me with the opportunity to review this significant piece of work. The authors have effectively tackled a crucial gap in the existing literature, specifically the modeling of access to health services in urban areas. While conventional methods like network analysis or least-cost path algorithms have proven beneficial in assessing access to care in rural regions, they fall short in capturing the complexity of travel time in urban settings. These approaches tend to underestimate actual travel time and may wrongly imply that access is not a concern in urban areas.

In this paper, the authors utilize Google data to estimate the median travel time for women of childbearing age in Nigerian cities. Their findings shed light on the overlooked issue of travel time to Comprehensive Emergency Obstetric and Neonatal Care (CEmONC) in urban areas, particularly impacting women in low-wealth quintiles. I commend the authors for their important contribution to the literature and for addressing this gap.

Response: Thank you very much for your kind commendation and indeed your detailed review. We have addressed all the points that you make below.

Moving forward, I would encourage the authors to consider further strengthening four main points:

1. Population distribution: The authors mention the use of a constrained population distribution layer specifically created for women of childbearing age. However, the process of constructing this layer is not well explained. It is worth noting that WorldPop typically does not generate constrained layers for sub-groups like this, and while pregnancy data is available at a 1km² resolution, it is unconstrained. Therefore, it would be beneficial for the authors to provide further clarification within the text regarding how they obtained or produced this specific layer. Especially because population data can cause large uncertainty in accessibility estimates (Hierink, et al. 2022)

Response: Thank you. We have clarified that the constrained age-sex-specific population dataset made available by Worldpop (<https://hub.worldpop.org/geodata/summary?id=50493>) was used instead of the pregnancy dataset. We limited the analysis to women aged between 15 and 49 years and did not perform any further computation to this layer. We have edited the manuscript to clarify this point.

2. S2 cells: The paper should explain the meaning of S2 cells a bit earlier. It is now explained in the sub-heading "Computing travel time and geographical coverage" but this should come in the subheading "Relative wealth" for people that do not know the Google data format.

Response: Thank you for picking this up. We have restructured the presentation of the methods to introduce S2 cells earlier.

3. Rural areas: In the discussion the authors do not mention whether this method can also be useful in rural areas or whether more classical methods are still important. Can the authors shine light on this?

Response: We appreciate your interest in rural areas and the relevance of methods you refer to as "more classical" for those settings; however, we believe that any mention of this in this paper will detract readers from the urban focus of this paper. Indeed, the current study is focused on Google's high applicability in settings where -

1. Google has good road data coverage,

2. Google's presence is high,
3. Assumed speed (basis of classical methods) is less likely to hold true in reality (e.g., due to traffic), because of the evidence base around Google out-performing classical methods published earlier under such settings/conditions.

We have an upcoming publication that compares the applicability of the Google API that we have used in this study and one of the classical methods, AccessMod, for use in urban and rural settings. Please feel free to reach out to inquire about this paper and we be happy to share with you.

4. Transferability: Can the authors add some information whether the travel times observed in Nigeria and the patterns found in the google data can be transferred to urban settings in other countries?

Response: We have included a reflection on this point in our concluding paragraph, writing, "The intra-urban inequitable patterns observed in this study might not be unique to Nigeria. In many other African cities that are set up in similar ways, health service provision, distribution, and their spatial relationship with the characteristics of communities remain work for future research." Scale up to assess transferability to other cities is work that our group is championing at the moment.

Minor points:

1. The following sentence in the introduction is a bit unclear: "However, there are ongoing debates about the adequacy of this travel time benchmark, as emerging evidence in urban African settings shows that women with pregnancy complications who reach care even within an hour of travel have significantly poorer outcomes." Poorer outcomes than who? Compared to rural populations?

Response: Compared to those who travelled less than 10 minutes. We have rephrased this sentence to read as, "However, there are ongoing debates about the adequacy of this travel time benchmark, as emerging evidence in urban African settings shows that women with pregnancy complications who reach care between 10 minutes and an hour of travel have significantly poorer outcomes than those who travel less than 10 minutes to care".

Here is the reference: Banke-Thomas, A. et al. Travel of pregnant women in emergency situations to hospital and maternal mortality in Lagos, Nigeria: a retrospective cohort study. *BMJ Glob Health* 7, e008604 (2022).

2. Figure 1 and Figure 2 could be merged and be different panels A, B, C, D

Response: Thank you for this comment. We have merged Figure 1 and Figure 2 and labelled them as different panels A, B, C, D.

3. In the results under Figure 2 rephrase: "Lagos remained with the widest inequality gap driven by many reachable facilities from Q5 S2 cells (n=263)". I would rephrase the sentence the other way around: "Lagos remained with the widest inequality gap driven by limited facilities available for the lowest wealth quintiles".

Response: Thank you for this recommendation. We have rephrased, as follows, "Lagos remained with the widest inequality gap, where 4 and 293 facilities were within reach under 30 minutes for the lowest and highest wealth quintiles, respectively".

Reviewer #2:

General comment - The manuscript reports a very interesting study with a novel methodology to fill the gaps in previous studies examining healthcare access and utilization issues, particularly among

women seeking obstetric care. The paper is well-written and the illustrations are adequate. A few areas I would want the authors to look at are as follow;

Response: Thank you very much for your kind commendation and indeed your detailed review. We have addressed all the points that you make below.

1. The term 'socioeconomic status' is quite a broad one and encompasses other indices beside wealth. I would suggest the authors stick with the wealth index they employed in the title or add a sentence in the abstract and introduction on using wealth index as a corollary of socioeconomic status in the study, instead of using both interchangeably.

Response: Thank you for this recommendation. We have used wealth index consistently across our manuscript. We have included a sentence to show the link between both terms, Writing, “Robust research to show relationships between travel time to care needs to be able to reflect closer-to-reality pathways to care and be linkable to spatially represented wealth index data of the population, which is commonly used as a proxy for socio-economic status in low- and middle-income country (LMIC) settings”.

2. The sentence 'In countries with a high maternal and perinatal morbidity and mortality burden, including many African countries like Nigeria...' in the first paragraph of the Introduction section would read better if buttressed with some current data on maternal and perinatal morbidity and mortality for context.

Response: We have included some current data on maternal and perinatal morbidity and mortality, writing, “In countries with a high maternal and perinatal morbidity and mortality burden, including many African countries like Nigeria, which contributes as much as a third of the 282,000 maternal deaths that occur annually worldwide...”.

3. The relationship between socioeconomic status and access to healthcare services is only mentioned in passing in the introductory section. The authors expanded elaborately on the aspects about travel time and access but did not do same to socioeconomic status in justifying the study objective.

Response: Thank you for picking this up. We have now added that, “However, her ability to access care can also be influenced by her socio-economic status” and cited a relevant literature to buttress the point.

4. Just like when using a DHS dataset for a study of this nature, the authors need to creatively explain the methodology in computing the relative wealth index. It is not sufficient to assume that because the data is in the public domain, everyone understands the metadata and how the index was generated. the RWI is a major explanatory variable in the study and should be amplified accordingly.

Response: We already included some detail on how RWI is computed before, but to address this comment, we have now expanded this section. The available details on methodology can be seen here:

<https://dataforgood.facebook.com/dfg/tools/relative-wealth-index#methodology>

5. The discussion section looks more like a combination of an extension and elaborated results section, a further justification of the methods employed, and a limitation to the study. Except if conforming to the journal's authors guidelines, this section should be unbundled.

Response: This is very much in line with the journal’s guidelines for the discussion section and aligned with style of other submissions to the journal. Please see example:

<https://www.nature.com/articles/s43856-023-00356-z>

6. Specific to the discussion, one would have loved to see how the study findings actually deviate / align with previous works and / or expand on previous studies that examined similar objectives but did not employ the methods used here. This will accentuate the shortcomings in those studies and further justify why this study was necessary in filling the gaps.

Response: We agree that this would have been useful, which is why in writing the discussion, we highlighted evidence from existing relevant literature to compare with our findings. Indeed, intra-urban gap in geographical access to healthcare is still fairly under-studied currently, and to our knowledge, none has used "big data". A few studies exploring the topic have been identified and referred to in the Discussion (e.g., Ref #24 and #41).

We thank you for your interest in our study and our group certainly aims to continue to push the frontier.

.

Reviewers' comments:

Reviewer #1 (Remarks to the Author):

Thank you very much for sharing the revised version of this interesting manuscript. I congratulate the authors for the improvements made. My comments are mostly sufficiently addressed. I have one remaining comment. The authors describe in their rebuttal letter that they have used constrained WorldPop population data while the article reads that the WoCBA at 1km² in unconstrained format have been used. Which one was used? In case of the latter, I highly recommend the authors to indeed use the constrained age-constructed women of 15-49 years WorldPop layer.

I appreciate the author's offer to share the paper on the comparison of the Google-based approach and the AccessMod approach. I'd be very interested to see those differences.

I wish the authors the best with the finalization of the paper.

Reviewer #3 (Remarks to the Author):

I think this is a very nice and thorough paper that should be published without major revision. The description of the methods are very detailed and easily meet the bar of replicability. The figures are beautiful, clear and coherent across the results section. Bravo.

My only substantive comment is that you do not have any policy takeaways from your analysis in the discussion. In the abstract, you say: "Geographical accessibility must be prioritised, more so for the particularly vulnerable populations living in urban settings." But what does this mean and how does your study indicate what should be done? Should more facilities be built in suburban areas? Should there be transportation vouchers or emergency ambulances available for obstetric emergencies? How should a policy-maker think about prioritizing suburban areas versus rural areas in terms of these policies?

Reviewer #4 (Remarks to the Author):

Please note that I am responding in place of Reviewer #2, who was unavailable for this round. I find that all of Reviewer #2's comments were adequately addressed in the response. Regarding comments #5 and #6 from the first round of review, the authors use the Discussion section to contextualize the findings of this study in the body of related research, in line with the expectations of this journal.

The authors have carefully limited their findings to the Relative Wealth Index and have noted the limitations of using this index to represent socio-economic status more broadly. The authors have also expanded their description of the Relative Wealth Index in the Methods section. The detail provided is reasonable given that the Index is described in full in another paper. My only request is that the authors directly cite the web page they listed in their response (<https://dataforgood.facebook.com/dfg/tools/relative-wealth-index>) in addition to their current citation for the Relative Wealth Index (#39).

Reviewers' comments:

Reviewer #1:

Reviewer #1 (Remarks to the Author):

Thank you very much for sharing the revised version of this interesting manuscript. I congratulate the authors for the improvements made. My comments are mostly sufficiently addressed. I have one remaining comment. The authors describe in their rebuttal letter that they have used constrained WorldPop population data while the article reads that the WoCBA at 1km² in unconstrained format have been used. Which one was used? In case of the latter, I highly recommend the authors to indeed use the constrained age-constructed women of 15-49 years WorldPop layer.

I appreciate the author's offer to share the paper on the comparison of the Google-based approach and the AccessMod approach. I'd be very interested to see those differences.

I wish the authors the best with the finalization of the paper.

Response: Thank you very much for your review and your very helpful comments. We are glad to read that we were able to address all your comments. Indeed, we used the constrained WorldPop population of WoCBA and appear to have not previously made the change in the manuscript. We have corrected this now and updated the reference accordingly.

Reviewer #3 :

I think this is a very nice and thorough paper that should be published without major revision. The description of the methods are very detailed and easily meet the bar of replicability. The figures are beautiful, clear and coherent across the results section. Bravo.

My only substantive comment is that you do not have any policy takeaways from your analysis in the discussion. In the abstract, you say: "Geographical accessibility must be prioritised, more so for the particularly vulnerable populations living in urban settings." But what does this mean and how does your study indicate what should be done? Should more facilities be built in suburban areas? Should there be transportation vouchers or emergency ambulances available for obstetric emergencies? How should a policy-maker think about prioritizing suburban areas versus rural areas in terms of these policies?

Response: Thank you very much for your kind and positive review. We are indeed grateful. Your nudge to flag potential interventions to address identified gaps is an essential point to make. We have now added in our Discussion that "Specific interventions to address observed inequities in geographical accessibility will vary depending on local context. However, such interventions must be pro-poor and could include construction of functional health facilities and strategic stationing of ambulances in suburbs and transport support provided to poor pregnant women to access care in an emergency" (Page 17, Paragraph 1). We are constrained by the 250-word limit of the abstract but have also flipped the conclusion there to read "Pro-poor interventions addressing geographical accessibility to CEmOC are needed to address observed inequities in urban settings". Thank you once again for this comment.

Reviewer #4 :

Please note that I am responding in place of Reviewer #2, who was unavailable for this round. I find that all of Reviewer #2's comments were adequately addressed in the response. Regarding comments

#5 and #6 from the first round of review, the authors use the Discussion section to contextualize the findings of this study in the body of related research, in line with the expectations of this journal.

The authors have carefully limited their findings to the Relative Wealth Index and have noted the limitations of using this index to represent socio-economic status more broadly. The authors have also expanded their description of the Relative Wealth Index in the Methods section. The detail provided is reasonable given that the Index is described in full in another paper. My only request is that the authors directly cite the web page they listed in their response (<https://dataforgood.facebook.com/dfg/tools/relative-wealth-index>) in addition to their current citation for the Relative Wealth Index (#39).

Response: Thank you for jumping in to help with this review. Much appreciated. We see the value in citing the website also and have now done this as advised (See new Ref 39).